# Water Stress Alters Physiological, Spectral, and Agronomic Indexes of Wheat Genotypes

**DOI:** 10.3390/plants12203571

**Published:** 2023-10-14

**Authors:** Cássio Jardim Tavares, Walter Quadros Ribeiro Junior, Maria Lucrécia Gerosa Ramos, Lucas Felisberto Pereira, Onno Muller, Raphael Augusto das Chagas Noqueli Casari, Carlos Antonio Ferreira de Sousa, Anderson Rodrigo da Silva

**Affiliations:** 1Federal Institute Goiano, Campus Cristalina (IF Goiano), Cristalina 73850-000, GO, Brazil; cassio.tavares@ifgoiano.edu.br; 2Brazilian Agricultural Research Corporation—(EMBRAPA Cerrados), Planaltina 73310-970, DF, Brazil; 3Faculty of Agronomy and Veterinary Medicine, University of Brasília, Brasília 70910-900, DF, Brazil; 4Federal Institute Goiano, Campus Posse (IF Goiano), Posse 73900-000, GO, Brazil; lucas.felisberto@ifgiano.edu.br; 5 Institute for Bio-and Geosciences, IBG-2: Plant Sciences, Forschungszentrum Jülich GmbH, 52428 Jülich, Germany; o.muller@fz-juelich.de; 6Institute of Geociences, University of Brasília, Brasília 70910970, DF, Brazil; casari.raphael@gmail.com; 7Brazilian Agricultural Research Corporation—(EMBRAPA Meio-Norte), Teresina 64008-780, PI, Brazil; carlos.antonio@embrapa.br; 8Federal Institute Goiano, Campus Urutaí (IF Goiano), Urutaí 75790-000, GO, Brazil; anderson.silva@ifgoiano.edu.br

**Keywords:** automation, *Triticum aestivum*, gas exchange, drought tolerance, high throughput, Cerrado

## Abstract

Selecting drought-tolerant and more water-efficient wheat genotypes is a research priority, specifically in regions with irregular rainfall or areas where climate change is expected to result in reduced water availability. The objective of this work was to use high-throughput measurements with morphophysiological traits to characterize wheat genotypes in relation to water stress. Field experiments were conducted from May to September 2018 and 2019, using a sprinkler bar irrigation system to control water availability to eighteen wheat genotypes: BRS 254; BRS 264; CPAC 01019; CPAC 01047; CPAC 07258; CPAC 08318; CPAC 9110; BRS 394 (irrigated biotypes), and Aliança; BR 18_Terena; BRS 404; MGS Brilhante; PF 020037; PF 020062; PF 120337; PF 100368; PF 080492; and TBIO Sintonia (rainfed biotypes). The water regimes varied from 22 to 100% of the crop evapotranspiration replacement. Water stress negatively affected gas exchange, vegetation indices, and grain yield. High throughput variables TCARI, NDVI, OSAVI, SAVI, PRI, NDRE, and GNDVI had higher yield and morphophysiological measurement correlations. The drought resistance index indicated that genotypes Aliança, BRS 254, BRS 404, CPAC 01019, PF 020062, and PF 080492 were more drought tolerant.

## 1. Introduction

Brazil has an annual demand of about 12 million tons of wheat grains, producing less than half of its needs and importing the other part [1]. Therefore, looking for alternatives to increase crop productivity and include new areas to reduce foreign dependence and expenditure on wheat imports is important.

Cerrado is a vast savanna biome in the Brazilian highlands and has been an alternative for rainfed or irrigated wheat cultivation [2], as it has favorable climatic and soil conditions. During winter in the Cerrado region, wheat has high yields under irrigation conditions and should be efficient in water use. In contrast, when planting wheat in the off-season, during the summer, the main limitation is dry periods, which require drought-tolerant plants [2].

Wheat cultivation during the summer (early crop) depends on the previous crop cycle, considering drought escape, water availability, drought-tolerant cultivars, and diseases such as blast (*Pyricularia oryzae*), which are the main limiting factors for wheat cultivation in the off-season [3].

However, in rainfed (off-season) cultivation, irregular rainfall and prolonged dry periods (dry spells) [4] are common in this region. Among the environmental changes that affect crop development, the ones caused by the water deficit, which intensified in recent years due to global warming [5,6], stand out. 

A reduction in precipitation in the critical stages of crop development has been reported, especially in the Brazilian Cerrado, one of the largest grain producers in the country [7], with drastic reductions in crop productivity [8]. In this scenario, understanding the mechanisms plants use to tolerate water stress is crucial to identifying wheat cultivars with tolerance and/or water-use efficiency, consequently decreasing the productivity and grain quality losses of these crops.

Drought can be minimized through proper management [9] and genetic tolerance. To unravel genetic tolerance, it is necessary to identify genotypes that minimally reduce their yield potential and grain quality under conditions of low water availability [1].

The survival of plants under drought depends on several mechanisms acting simultaneously: reduction of the growing cycle, stomatal opening and transpiration rate, and the development of antioxidants that maintain osmotic adjustments at the tissue level [10]. Consequently, morphophysiological and crop yield changes may occur [11].

Because of the water deficit effects on plants, genotypes tolerant to water stress can be identified by combining several variables, namely drought tolerance indices [2], water-use efficiency [12], productivity components [13], physiological index [14,15], and vegetation (spectral) indices [16,17], and these variables may be correlated with grain yield and quality [18]. The effect of stomatal closure is a reduction in photosynthesis, but screening based on photosynthetic parameters or stomatal conductance measurements is generally slow [19]. 

New technologies and methodologies that measure these variables should offer advantages in relation to yield levels, applicability in field conditions (in various climatic conditions), and speed in obtaining this information. The use of tools that provide a rapid assessment and are non-invasive and destructive at any stage of crop development and with high precision on the characteristics of plants in relation to abiotic and biotic factors has been used in the studies of the interaction plant environment [20].

A phenotyping platform for field drought tolerance was developed and used at Embrapa Cerrados as described by Jayme-Oliveira [21] for several annual and perennial species. Automated watering systems, often integrated with high-throughput phenotyping tools, are critical to experimentation, as a manual water application is imprecise and labor-intensive [22]. 

Among these tools, using sensors in agriculture can provide valuable information on plant physiology under water stress by detecting changes related to plant structure, pigments, and photosynthetic efficiency [23].

Using sensors coupled on land platforms [24] or unmanned aerial vehicles [25] has the advantage of a rapid, non-destructive evaluation with high precision. However, there is a need to validate the efficiency of these tools. Among these sensors, multispectral cameras are coupled in unmanned vehicles, which can be used to select cultivars adapted to different conditions [26], an indispensable tool in breeding programs.

Thus, multispectral sensors that capture wavelengths in the visible and near-infrared range of the electromagnetic spectrum allow the calculation of various vegetation indices [20] and can be used for rapid and non-destructive selection of drought-tolerant wheat genotypes [27] with high precision [28]. Multispectral traits derived from NIR, red, and green bands showed a strong relationship with wheat biomass, water-use efficiency, photosynthesis, and grain yield. The study of morphophysiological, spectral, and productivity changes due to water stress in wheat genotypes is important for selecting cultivars that are more tolerant to water stress [17]. 

Therefore, the characterization of genotypes tolerant to water stress in rainfed cultivation and greater water-use efficiency in irrigated cultivation under Cerrado [2] conditions is important to increase grain yield and quality. This work hypothesizes that non-destructive physiological responses of wheat, as well as high throughput vegetative indices, are related to soil water availability and can be used to discriminate tolerant genotypes to water stress. Thus, this work aimed to validate the use of sensors as a tool for selecting wheat genotypes for drought tolerance through morphophysiological and agronomic assessments under field conditions.

## 2. Results

### Variable Contributions in the Multivariate Response

Data were submitted to joint multivariate analysis of variance, and significant differences were found for the sources of variation genotypes (*p* < 0.01), water regime (*p* < 0.01), year of cultivation (*p* < 0.01), and the interaction genotypes x water regime (*p* < 0.01) (Table 1). However, there was no significant (*p* > 0.05) effect of the interaction genotype × water regime × cropping year (*p* = 1). The significance of genotype × water regime interaction shows that genotypes respond differently to water availability.

After a joint multivariate statistical analysis, the wheat genotypes were grouped in Group 1: Aliança, BRS 254, BRS 404, CPAC 01019, PF 020062, and PF 080492; Group 2: BR 18_Terena, MGS Brilhante, PF 020037, and PF 120337; Group 3: BRS 264, BRS 394, CPAC 01047, CPAC 07258, CPAC 8318, CPAC 9110, PF 100368, and TBIO Sintonia. The average values of the variables analyzed in the wheat crop according to the groups of genotypes and water regimes in 2018 and 2019 are presented in Table 2 and Table 3. In general, changes in water availability affected the morphophysiological and spectral characteristics and crop yield, with different levels in the genotypes studied.

In Figure 1, three groups of genotypes were identified. Group 1 with six genotypes (33.33%) (Aliança; BRS 254; BRS 404; CPAC 01019; PF 020062; and PF 080492); Group 2 has four genotypes (22.22%) (BR 18_Terena; MGS Brilhante; PF 020037; and PF 120337) and Group 3 has eight genotypes (44.45%) (BRS 264; BRS 394; CPAC 01047; CPAC 07258; CPAC 8318; CPAC 9110; PF 100368; and TBIO Sintonia).

The genotypes in Group 1 had a higher drought resistance index (DRI = 1.01) than the genotypes in Group 3 (DRI = 0.93), with intermediate values in Group 2 (DRI = 0.95) (Figure 2). Vegetative indices accounted for 97% of the total divergence observed among genotypes under water stress (Table 3). Indices based on the near-infrared band (OSAVI (33%), SAVI (33%), NDRE (11%), DVI (11%), GNDVI (3%), and NDVI (1%)) were most strongly associated with the differentiation of genotype groups. In addition, they were among the variables most affected by water stress.

Figure 2 shows the results of DRI (drought resistance index) for each group of wheat genotypes.

The singular value decomposition of means of combinations of water regime levels and genotype groups retained 86% of the total variability in the first principal coordinate (Figure 3). However, there is a strong contrast in responses among genotype groups that received WR1 (22% of CET replacement) compared to the same groups with WR4 (100% of CET replacement). This means that the selection process must be carried out exactly under the conditions at the field, either under irrigation or under rainfed conditions.

The variable gas exchange (*A*, *g_s_* and *E*) and vegetative indices (NDVI, GNDVI, GRVI, DVI, NDRE, SAVI, PRI, OSAVI, and TCARI) are correlated (Figure 3 and Figure 4), and the higher correlations are between *A* and NDGI (0.78), *gs* and RVI (0.73), and *E* and NDGI (0.70).

The variables iWUE, WUE, and Rm/Sm ratio positively correlate. In contrast, WUE and net CO_2_ assimilation (*A*) negatively correlated (Figure 3 and Figure 4). There are correlations between vegetative and gas exchange indices as they formed sharp angles between these variables, showing a correlation with grain yield, which should be the main criteria for the selection because yield is the most important variable.

All IVs except TCARI/OSAVI (TO) have a strong positive correlation (>0.6, *p* < 0.05) with grain yield, photosynthetic rate, stomatal conductance, and transpiration (Figure 4). We observed correlations of 0.33 of total chlorophyll and 0.07 of root mass with grain yield, respectively. Mass of a thousand grains and grain yield are correlated (0.55). Negative correlations above 0.5 occur between these variables with WUE, iWUE, TO, and ratio chlorophyll a to chlorophyll b. Positive correlations are observed between the WUE and iWUE variables (0.3, *p* < 0.05) (Figure 4).

Photosynthesis (*A*) and stomatal conductance (*g_s_*) showed a high correlation with productivity (0.62 and 0.63, respectively), but these evaluations can not be performed on a large scale. On the other hand, the vegetative indices TCARI, NDVI, OSAVI, SAVI, PRI, NDRE, and GNDVI showed the highest correlation with productivity (>0.7, *p* < 0.05). They can be useful in breeding programs (Figure 4) as they can be evaluated on a large scale. The vegetation indices were generally correlated, and not all of them need to be assessed for selecting wheat genotypes.

Variables related to vegetative indices, gas exchange, chlorophyll, MTG, GY, WUE, and iWUE contents were the most important to differentiate genotype groups and water regime levels (Figure 3), as they have weights (length of arrows (≅1.0)) greater, indicating that these variables can be used to select wheat genotypes more productive under water stress and are also important components in the process of selecting more productive materials. Root mass (0–20 cm) had a low weight (0.25) for differentiating water regimes and genotype groups.

Without water stress, genotype groups with WR4 (100% of CET replacement) are mostly on the right side of the biplot, meaning that they present higher vegetative indices and photosynthetic activity, promoting higher grain yields (Figure 3).

The groups of genotypes showed linear positive responses of the latent variable (principal coordinate 1) (Table 5) as a function of the water regime. Additionally, we noticed that Group 3 has a response rate (slope = 0.0326) 18% higher on average. In contrast, the other two groups have similar slopes, probably because the genotypes grouped in three are classified as biotypes of irrigated crops, except for PF 100368 and TBIO Sintonia. The changes in water availability lead to significant changes in all traits according to the weights (horizontal arrow length) in the principal coordinate 1 (Figure 3).

## 3. Discussion

This paper studied the effect of four water regimes in twenty-seven variables and eighteen wheat genotypes. Through the joint multivariate analysis, we could separate the wheat genotypes into three groups (Table 2 and Table 3) through genetic distance studies. This indicates that genetic variability in wheat genotypes among the studied variables is essential for conducting genetic distance studies (Figure 1). In addition, different responses to water regimes also occur among irrigated and rainfed wheat biotypes and other species, such as triticale and common beans [30]. 

The most similar genotypes were Aliança and BRS 254, and those that differed the most from PF 020037 and PF 120337. Aliança is a rainfed genotype and, in a previous study, was considered the most drought-tolerant genotype in severe water stress [2]. It is interesting to consider that rainfed and irrigated biotypes have distinct breeding programs conducted at different periods of the year. The irrigated biotype is cultivated in a more favorable climatic period for the species due to the cold at night and irrigated, which means a higher yield. In the target region (Cerrado), light is not a limiting factor in both planting periods.

Tavares et al. [31] studied six soybean genotypes in the Brazilian Cerrado region during the dry season and showed genotypes adapted to different water regimes.

The study carried out by Soares et al. [2] confirms the clustering results from this work, in which rainfed genotypes (Aliança, MGS Brilhante, and BRS 404) had higher DRI than irrigated genotypes (BRS 394, BRS 254, and BRS 264) after two years of cultivation. This was expected since the genotypes classified in Group 1, except for CPAC 01019 and BRS 254, are rainfed biotypes. In contrast, the members of Group 3, except for genotypes PF 100368 and TBIO Sintonia, are biotypes for irrigated cultivation, which generally have a low DRI since they were developed for environments without water restriction. The wheat genotypes with higher DRI can be selected as a reference in breeding programs as they were more tolerant to water stress (Figure 2). 

Using vegetative indices through cameras coupled in VANTS is a quick and non-invasive complementary method or can be replaced with the traditional selection of genotypes more tolerant to water stress [32,33,34].

Vegetative indices based on near-infra-red bands could separate the groups of wheat genotypes and were more affected by water stress, indicating the possibility of using vegetative indices as a quick and non-destructive way to differentiate and select wheat genotypes. Overall, these results are consistent with data from other authors who have found that NIR-based spectral indices have high accuracy and efficiency in selecting more productive wheat genotypes under water stress [28,35]. In contrast, indices based on the visible range correlate more strongly with plant growth [17,36].

In addition, spectral indices collected via cameras coupled to aircraft consistently show a stronger association with grain yield than indices obtained using proximal methods, indicating greater precision, speed, and scale gains [28].

Positive correlations of the NDVI, GNDVI, TCARI, and OSAVI vegetation indices and negative correlations between the TCARI/OSAVI index with shoot biomass and grain yield in wheat are reported by Frels et al. [37]. The TCARI/OSAVI index is very sensitive to chlorophyll variation and resistant to the effect of soil reflectance and non-photosynthetic matter, distorting the response in relation to the other index [38]. The non-destructive, simple, and fast measurement characteristics of PRI and NDRE constitute an important advantage over the physiological parameters (contents of photosynthetic pigments, chlorophyll fluorescence, and gas exchange rate). They will make it useful in stress detection, especially under severe levels or late stages of heat and water stress [39].

Genotypes under stressed conditions (WR1: 22% CET replacement) have higher values of WUE, iWUE, and TO (TCARI/OSAVI) (Table 2 and Table 3) because there was less water lost, and TO, for instance, is a spectral predictor that is sensitive to chlorophyll variation [40].

The roots contribute only 10 to 20% of the total plant weight, but a well-developed root system is essential for the nutrition of nutrients and water and, therefore, for the growth and final yield of the plants [41] under conditions of limited water availability during vegetative growth, such as plants seeking to increase root volume to improve water uptake and avoid yield losses.

Water availability promoted significant alterations in all traits, according to the weights (horizontal arrow length) in the principal coordinate 1 (Figure 3). These observations confirm the findings of other authors who reported that under water deficit conditions, water relations and plant metabolism are impaired [15,42]. Photosynthesis can be affected by Rubisco activity and limited by chloroplast CO_2_ concentration, i.e., under water deficit conditions, plants close stomata to prevent transpiration, which reduces photosynthesis by lowering CO_2_ influx [14].

In addition to direct limitation, water stress can cause morphological changes in plants, such as changes in the organization of chloroplasts and the number of pigments, especially chlorophylls and xanthophylls, which consume energy and lead to lower grain yields [11,14]. Additionally, water stress reduces the number and length of the ears of wheat [2]. Abscisic acid (ABA) generally regulates and stimulates reactions such as stomata closure and maintaining water balance, and it stimulates transcription and activities of antioxidant enzymes under water deficit conditions [43].

In addition to drought tolerance, escape may occur because of some mechanism that prevents drought, including reduced leaf area, chlorophyll content, number of tillering, plant height, and stomatal conductance to avoid water loss; increased root length and root-to-shoot ratio, probably to increase water uptake capacity; and increased proline content in leaves and reduce cycling [44]. Additionally, water stress reduced the plant cycle in our experiments, similar to Tavares et al. [31] working with soybeans.

Other studies show that water stress alters the morphology of the root system of wheat plants. For instance, changes in the root angle, primary root length, number of lateral roots [45], mean root diameter [46], and root mass [27] have been reported in the literature.

In the present study, with WR1 (22% CET replacement), a 31% reduction in shoot mass was observed in relation to the non-stressed water regime (WR4—100% CET replacement) (Table 3). A decrease in the shoot mass ratio and root mass suggests that wheat plants increase the root system to absorb more water. Furthermore, under a water deficit, wheat develops nodal roots with a narrower angle, which tend to grow deeper than those with a larger root angle and increase root density and decrease diameter; consequently, roots achieve deeper layers of soil [47].

The water regimes most affected the grain yield, net assimilation rate of CO_2_, water-use efficiency, and NDVI (Figure 3). The groups of genotypes showed exponential responses of grain yield and net assimilation of CO_2_ and a logarithm response of NDVI, water-use efficiency, and intrinsic water-use efficiency (Table 5).

Increasing water availability increased NDVI and photosynthetic activity, promoting higher grain yield. On a descending scale, we observed that genotypes of Groups 2 and 3 respond faster to water availability than genotypes of Group 1 for the variables grain yield, net CO_2_ assimilation, and NDVI.

There is greater water-use efficiency under lower water availability. However, this efficiency is reduced more quickly in the genotypes of Groups 3 and 2 than in the genotypes of Group 1.

Soares et al. [2] also obtained differences between wheat genotypes for drought tolerance. The number of ears, the mass of a thousand grains, grain yield, net CO_2_ assimilation, and hectoliter weight are important characteristics for selecting more productive genotypes. In addition, the variables DRI and water-use efficiency are important to classify genotypes for drought tolerance.

Water-use efficiency is a variable related to the ability of a plant to produce grains with a lower amount of water. Water-use efficiency by wheat plants increases under conditions of moderate water stress and decreases under stress-free conditions [11,12]. Using genotypes with greater water-use efficiency [48], especially for species with high water demand cultivated in rainfed agriculture [49], reduces crop losses in areas with water shortages.

## 4. Materials and Methods

### 4.1. Experimental Design and Conducting the Experiment

The study was carried out in the wheat cultivation areas in Planaltina, DF, Brazil (15°35′30″ S and 47°42′30″ W, an altitude of 1006 m) between May and September 2018 and 2019. The climate is classified as Aw (Koeppen–Geiger)—tropical, with rainfall concentrated in summer (October to April) and the dry period during the winter (May to September). The annual rainfall is between 1200 and 1500 mm. Precipitation and air temperature data from the experimental area in 2018 and 2019 are presented in Figure 5 and were obtained from a meteorological station close to the experiment. The experimental area had been cultivated for the last four years with wheat under different water regimes in winter and fallow in summer.

The soil is classified as Oxisol [50]. Before the installation of the experiment, soil samples were collected at a depth of 0–20 cm, with the following physicochemical properties: pH (CaCl_2_) of 5.7, 11 mg dm^−3^ P (Mehlich–1), 186 mg dm^−3^, K 5.77 cmolc dm^−3^ Ca, 1.83 cmolc dm^−3^ Mg, 0.02 cmolc dm^−3^ Al, and 26.7 g kg^−1^ organic matter. The soil texture comprised 46, 10, and 44% of clay, silt, and sand, respectively.

The model fitted the soil water retention curve [51]. The following estimates were obtained: residual water content (θs) 0.0839 cm^3^ cm^−3^, saturated water content (θs) 0.5500 cm^3^ cm^−3^, and parameters α (1.892 kPa^−1^) and n (1.2390). The field capacity moisture was 0.3423 cm^3^ cm^−3^.

Wheat biotypes developed for rainfed and irrigated planting periods were used, with different breeding improvement programs for the Cerrado region. The criterion for choosing the tested genotypes was to use recent and traditionally released materials and elite genotypes arising from the recent breeding program. The experiment was conducted in winter because there is almost no precipitation, which makes it easier to obtain an irrigation gradient. 

The experimental design consisted of randomized blocks in a split-plot scheme with three replicates. The plots consisted of eighteen wheat genotypes: BRS 254, BRS 264, CPAC 01019, CPAC 01047, CPAC 07258, CPAC 08318, CPAC 9110, BRS 394 (irrigated biotypes) and Aliança, BR 18_Terena, BRS 404, MGS Brilhante, PF 020037, PF 020062, PF 120337, PF 100368, PF 080492, TBIO Sintonia (rainfed biotypes) all with similar cycle of approximately 110 days. Subplots were composed of four water regimes (WR). In 2018, 123.68 mm, 241.74 mm, 455.38 mm, and 562.2 mm were applied during the cycle, corresponding to WR1, WR2, WR3, and WR4, respectively. In 2019, 119.1 mm, 232.15 mm, 438.55 mm, and 541.43 mm were applied, corresponding to WR1, WR2, WR3, and WR4, respectively. In 2018 and 2019, the irrigation regimes used corresponded to 22%, 43%, 81%, and 100% of crop evapotranspiration (CET) replacement, respectively (Figure 6).

The rainfed biotypes were MGS Brilhante, Aliança, TBIO Sintonia, BR18_Terena (all traditional material), and BRS 404 (registered in 2014). The rainfed lines developed for the Cerrado region were PF020037 (strong wax formation on leaves and stems) and PF020062 (but without wax formation). In addition, the rainfed biotype PF080492 was adapted to the Brazilian Cerrado region [2]. The irrigated cultivars were BRS 254, BRS 264 (traditional materials), and BRS 394 (registered in 2014), and in the experimental phase for commercialization, the irrigated biotypes CPAC 01019, CPAC 01047, CPAC 07258, CPAC 08318, and CPAC 9110.

The highest-level irrigation was carried out according to the Cerrado Irrigation Monitoring Program [52], that is, by replacing evapotranspiration using regional agrometeorological indicators, soil texture, and date of total plant emergence date of the plants and estimating the reference evapotranspiration with based on the equation proposed by Penman-Monteith [53]. Irrigation was carried out approximately every five days, according to weather conditions and the phenological phase of the plants. The amount of water applied in each irrigation was estimated by collectors placed parallel to the irrigation bar. 

Twenty days before planting, glyphosate was applied (1440 g.e.a ha^−1^). Seeding was done mechanically on 22 May 2018 and 23 May 2019 in a no-till system with 90 seeds per meter at a depth of 3 cm. Fertilization was carried in the furrows at 400 kg ha^−1^ of 04-30-16 (N, P_2_O_5_, and K_2_O), and at 25 days after wheat emergence (DAE), nitrogen was applied at a dose of 90 kg ha^−1^ N, as urea.

Furthermore, at 36 and 38 DAE, in 2018 and 2019, the growth reducer tranexamic-ethyl at a dose of 125 g ha^−1^ was applied in the first visible node and with the second palpable node. In addition, the herbicide metsulfuron-methyl was applied at 4 g ha^−1^ in both years for weed control on 15 DAE.

In both experiments, a homogeneous water layer was applied during the first 35 DAE at the tillering stage. An average of 150 mm of water was applied to obtain a homogeneous plant stand. After this period, the “line source” method was applied and modified by Jayme-Oliveira et al. [21]. The water regime (WR) was obtained by an irrigation bar (IrrigaBrasil model 36/42) 20 m wide on each side of the bar. The bar was connected to a self-propelled TurboMaq 75/GB) with controlled speed.

Each experimental unit consisted of a genotype with 18.0 m formed by eight cultivation rows spaced 0.17 m apart. Each water regime consisted of an experimental subunit 2.0 m in length formed by eight rows 0.17 m apart. The usable area consisted of the six central rows, omitting the margins and 2 m on each side.

### 4.2. Analyzed Variables

At the flowering stage (75 DAE), gas exchange and electron transport rate were evaluated from 8:30 am to 12:30 pm at an irradiance of 1200 μmol photons m^−2^s^−1^ and an external CO_2_ concentration (Ca) of 400 μmol mol^−1^. For each subplot, three fully expanded flag leaves were used to evaluate gas exchange using a portable open-flow gas exchange system (IRGA, model LI-6400xt LI-COR Inc., Lincoln, NE, USA). In addition, chlorophyll fluorescence and maximum quantum yield of photosystem II (Fv/Fm) were evaluated with a modulated portable fluorometer coupled to an IRGA. Evaluations were performed on dark-adapted leaves for at least three hours after 10:30 pm. During this period, initial fluorescence (F0), maximum fluorescence (Fm), and potential quantum yield of photosystem II (Fv/Fm = (F0 − Fm)/Fm)) were estimated [54].

Chlorophyll a and b content were measured with the ClorofiLOG portable chlorophyll meter (CFL-1030, model Falker – Falker Agricultural Automation, Porto Alegre, Brazil), which provides relative measurements (0 to 100) of total chlorophyll but is linearly correlated with total chlorophyll content [55].

Using a multispectral camera, model Micasense RedEdge, which captures images in five different spectral ranges, that is, range: 465–485 nm; range: 550–570 nm; range: 663–673 nm; range: 712–722 nm; and range: 820–860 nm, coupled to an unmanned aerial vehicle (UAV) with a rotary wing, images were taken to estimate the vegetation indices in the flowering phase of the crop. This camera features an optical resolution of 1280 × 960 pixels, with images recorded in RAW12 bits. The flight took place at a height of 45 m at 10 am. The reflectance maps were calculated by generating mosaics in the Pix4D Mapper software (v5.4.6, Pix4D, Lausanne, Switzerland).

The maps were processed in R software version 3.6.1 using the raster package, and the following vegetative indices were extracted as in Tavares et al. [31]: normalized difference vegetation index—NDVI; green normalized difference vegetation index—GNDVI; green–red vegetation index—GRVI; difference vegetation index—DVI; normalized difference red edge—NDRE; soil-adjusted vegetation index—SAVI; photochemical–physiological reflectance index—PRI; optimized soil-adjusted vegetation index—OSAVI; chlorophyll absorption and reflectance index—TCARI; TCARI/OSAVI—TO ratio.

Soil samples were collected to quantify the root mass of wheat genotypes during flowering. An auger with sharp edges, an internal diameter of 9.8 cm, and a length of 20 cm was used for root sampling. For each experimental subunit, 1508 cm^3^ of soil with roots was sampled, and the auger was inserted over the wheat crop line. The aerial part of the wheat plants corresponded to the area of the auger where the soil samples with roots (9.8 linear cm) were removed, cut close to the soil with a knife, and placed in a paper bag. The soil auger is made of stainless steel and has an iron shaft to facilitate rotation and application of force. This sampling procedure was adopted as in Ratke et al. [56].

The root samples with soil were kept in plastic bags, and the shoots in paper bags. The soil mass with roots was determined, and a subsample of 100 g was used to determine soil moisture content. Samples were stored in cold rooms at a temperature of −5 °C possible until the roots were separated from the soil to avoid loss of mass or drying the roots. Roots were separated from soil by washing with water three times in a 500 µm sieve and later separated from other organic materials. The roots and shoots were kept in a convection oven at 60 °C for 72 h to determine the dry matter. The results were expressed as root dry mass to soil mass ratio and root to shoot ratio.

At harvest, grain yield (GY) on the usable area of each experimental subunit and mass of thousand grains (MTG) was measured and standardized to the grain moisture content of 13% on a wet basis. Water-use efficiency (WUE) was calculated using the ratio of grain yield to crop water requirement [57]. The intrinsic water-use efficiency (iWUE) was calculated by the ratio between the net assimilation of CO_2_ and stomatal conductance (A/gs). The drought resistance index was calculated according to Fischer and Maurer [58].

### 4.3. Statistical Analysis

The data were subjected to joint multivariate analysis of variance by harvest based on singular value decomposition (SVD). Wheat genotypes were grouped based on the Mahalanobis distance using Ward’s method. The Mojena criterion [59] was used to define the cutoff point in the dendrogram, and the relative importance (proportion) of variables in the distance between genotypes was determined by the Singh [29] criterion. In both years, treatments (combinations of genotypes and water regime levels) were analyzed graphically in a biplot [60]. This allows visualizing the relationship between genotypes and treatments. A Pearson correlation analysis (*t*-test, *p* < 0.05) was performed with the residuals. The statistical analyses were performed using the R v3.6.1. software.

## 5. Conclusions

High-throughput and non-destructive vegetative indices based on the near-infrared band can be used to detect drought tolerance of wheat genotypes, as they are correlated with physiological and agronomic traits. The wheat genotypes can be clustered according to their responses to water stress. The higher drought resistance index genotypes are Aliança, BRS 254, BRS 404, CPAC 01019, PF 020062, and PF 080492. On the other hand, the genotypes BRS 264, BRS 394, CPAC 01047, CPAC 07258, CPAC 8318, CPAC 9110, PF 100368, and TBIO Sintonia are the most affected by water availability. There is less variability in wheat genotypes’ physiological, agronomic, and spectral responses when providing water based on 100% evapotranspiration.

## Figures and Tables

**Figure 1 plants-12-03571-f001:**
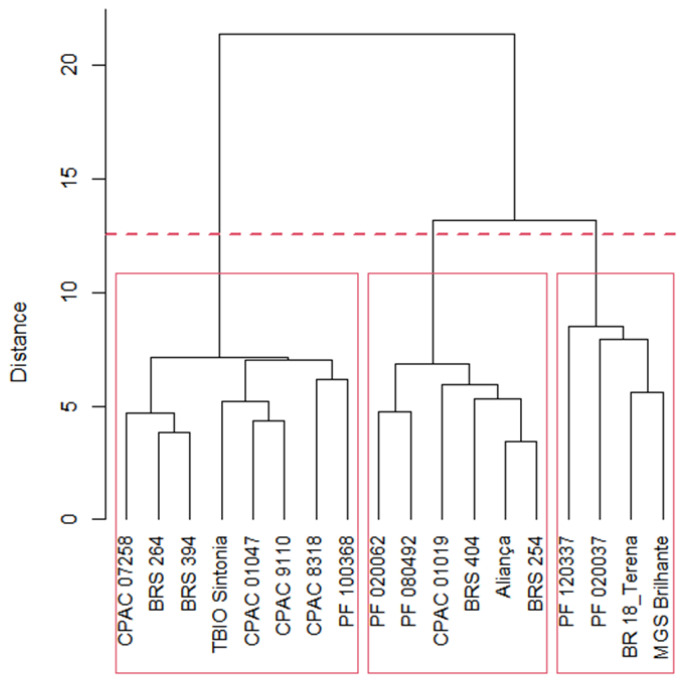
Dendrogram of eighteen wheat genotypes, based on Ward’s clustering using the generalized Mahalanobis distance. Group 1: Aliança, BRS 254, BRS 404, CPAC 01019, PF 020062, and PF 080492; Group 2: BR 18_Terena, MGS Brilhante, PF 020037, and PF 120337; Group 3: BRS 264, BRS 394, CPAC 01047, CPAC 07258, CPAC 8318, CPAC 9110, PF 100368, and TBIO Sintonia.

**Figure 2 plants-12-03571-f002:**
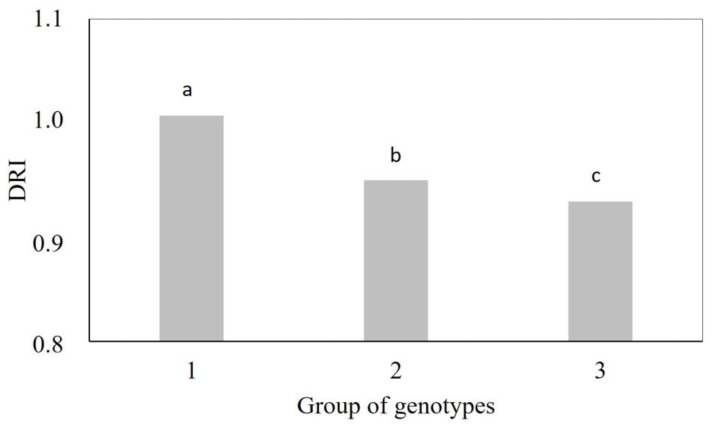
Drought resistance index (DRI) of groups of wheat genotypes. Group 1: Aliança, BRS 254, BRS 404, CPAC 01019, PF 020062, and PF 080492; Group 2: BR 18_Terena, MGS Brilhante, PF 020037, and PF 120337; Group 3: BRS 264, BRS 394, CPAC 01047, CPAC 07258, CPAC 8318, CPAC 9110, PF 100368, and TBIO Sintonia. Means followed by the same letter do not differ by the Tukey’s test (*p* < 0.05).

**Figure 3 plants-12-03571-f003:**
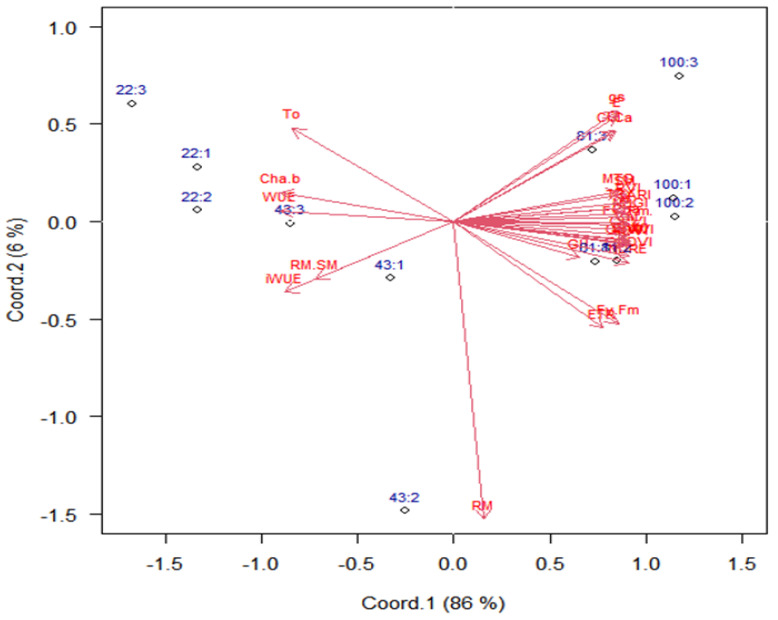
Biplot for scores of combined levels of water regimes (22, 43, 81, and 100% CET replacement, respectively) and wheat genotype groups (1, 2, and 3), based on the scores of principal coordinates from the variables: root mass (RM); shoot mass (SM); ratio RM/SM (RM.SM); grain humidity (GH); mass of thousand grain (MTG); grain yield (GY); water-use efficiency (WUE); intrinsic water-use efficiency (iWUE—*A*/*gs*: net assimilation of CO_2_/stomatal conductance); transformed chlorophyll absorption and reflectance index (TCARI); green red vegetation index (GRVI); normalized difference vegetation index (NDVI); optimized soil-adjusted vegetation index (OSAVI); soil-adjusted vegetation index (SAVI); photochemical–physiological reflectance index (PRI); green normalized difference vegetation index (GNDVI); red edge normalized difference (NDRE); difference vegetation index (DVI); net assimilation of CO_2_ (*A*); stomatal conductance (*gs*); transpiration (*E)*; internal CO_2_ concentrations (Ci); electron transport rate (ETR); maximum quantum yield of photosystem II (Fv/Fm); effective quantum yield of photosystem (Fv’/Fm’); chlorophyll a (Cha); chlorophyll b (Chb); ratio chlorophyll a/b (Cha.b). Group 1: Aliança, BRS 254, BRS 404, CPAC 01019, PF 020062, and PF 080492; Group 2: BR 18_Terena, MGS Brilhante, PF 020037, and PF 120337; Group 3: BRS 264, BRS 394, CPAC 01047, CPAC 07258, CPAC 8318, CPAC 9110, PF 100368, and TBIO Sintonia.

**Figure 4 plants-12-03571-f004:**
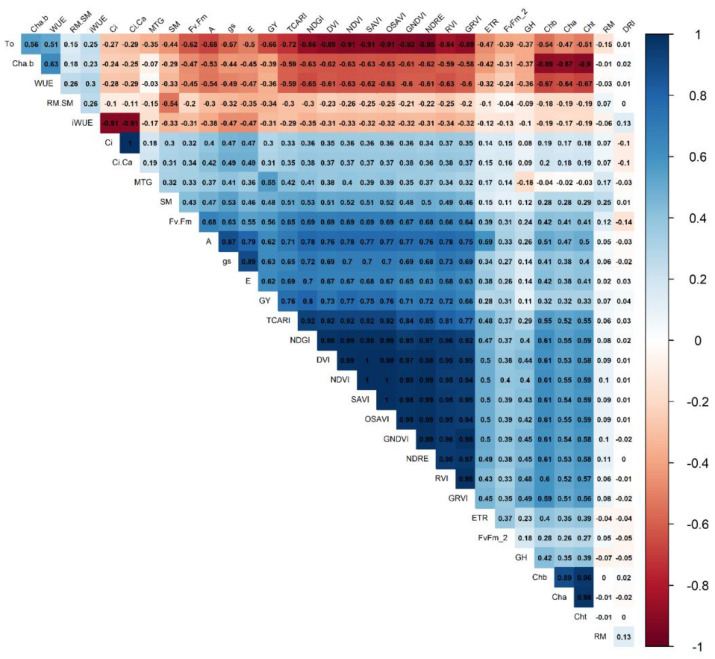
Pearson correlogram between variables representing physiological, spectral, and agronomic as a function of water regimes (22, 43, 81, and 100% CET replacement—WR1, WR2, WR3, and WR4, respectively) and wheat genotype groups (1, 2, and 3), based on the scores of principal coordinates from the variables: ratio RM/SM (RM.SM); ratio chlorophyll a/b (Cha.b); water-use efficiency (WUE); TCARI/OSAVI ratio (TO); intrinsic water-use efficiency (iWUE—A/gs: net assimilation of CO_2_/stomatal conductance); electron transport rate (ETR)); maximum quantum yield of photosystem II (Fv/Fm); grain humidity (GH); chlorophyll b (Chb); chlorophyll a (Cha); chlorophyll total (Cht); drought-resistance index (DRI); root mass (RM); internal CO_2_ concentrations (Ci); mass of thousand-grain (MTG); shoot mass (SM); effective quantum yield of photosystem (Fv’/Fm’); stomatal conductance (*gs*); transpiration (*E*); grain yield (GY); net assimilation of CO_2_ (*A*); the transformed chlorophyll absorption and reflectance index (TCARI); green red vegetation index (GRVI); normalized difference vegetation index (NDVI); optimized soil-adjusted vegetation index (OSAVI); soil-adjusted vegetation index (SAVI); photochemical–physiological reflectance index (PRI); green normalized difference vegetation index (GNDVI); red edge normalized difference (NDRE); difference vegetation index (DVI); normalized differential greenness index (NDGI); ratio vegetation index (RVI). Group 1: Aliança, BRS 254, BRS 404, CPAC 01019, PF 020062, and PF 080492; Group 2: BR 18_Terena, MGS Brilhante, PF 020037, and PF 120337; Group 3: BRS 264, BRS 394, CPAC 01047, CPAC 07258, CPAC 8318, CPAC 9110, PF 100368, and TBIO Sintonia.

**Figure 5 plants-12-03571-f005:**
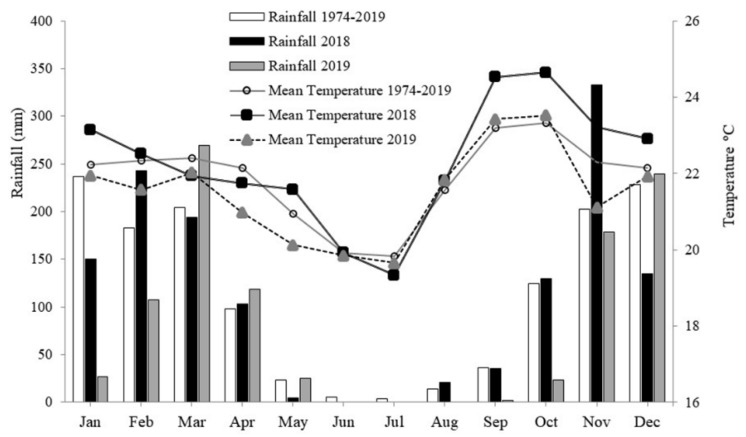
Precipitation and average temperature measured by an automatic weather station near the experiment in 2018 and 2019 and a historical series (1974–2019).

**Figure 6 plants-12-03571-f006:**
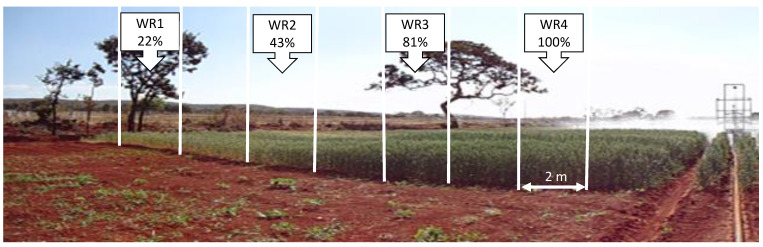
Graphical summary of experimental subunits (WR1, WR2, WR3 e WR4—22, 43, 81, 100% CET replacement, respectively) and the irrigation bar in wheat crop.

**Table 1 plants-12-03571-t001:** Summary of the multivariate analysis of variance (MANOVA) joined with all interactions.

Source	Df	Pillai	Approx F	Num Df	Den Df	Probability
Year	1	0.882	44.831	31	186	2.2 × 10^−16^ **
Genotypes	17	6.5224	4.056	527	3434	2.2 × 10^−16^ **
Water regime	3	2.6453	45.234	93	564	2.2 × 10^−16^ **
Year:Block	4	2.2584	7.906	124	756	2.2 × 10^−16^ **
Year:Genotypes	17	1.6662	0.708	527	3434	1 ^ns^
Year:Water regime	3	0.9002	2.600	93	564	7.026 × 10^−12^ **
Genotypes:Water regime	51	9.6958	1.928	1581	564	2.2 × 10^−16^ **
Year:Block:Genotypes	68	9.9413	1.500	2108	6699	2.2 × 10^−16^ **
Year:Genotypes:Water regime	51	1.8329	0.266	1581	6699	1 ^ns^
Residuals	216					

**: *p* < 0.01; ns: not significant.

**Table 2 plants-12-03571-t002:** Mean values of vegetative indices and wheat yield according to the groups of genotypes and water regimes in the 2018 and 2019 cropping years.

G^1/^	WR	Variables												
		NDVI	SAVI	PRI	DVI	GRVI	GNDVI	NDRE	TCARI	OSAVI	TO	GH	MTG	GY
1	22	0.27	0.18	0.1	1.81	3.09	0.5	0.12	0.12	0.22	0.54	10.5	32.97	2337
	43	0.45	0.28	0.16	3.17	3.89	0.58	0.25	0.14	0.36	0.4	10.8	35.19	3691
	81	0.62	0.38	0.21	5.58	4.7	0.63	0.34	0.18	0.49	0.38	11.3	37.56	5076
	100	0.65	0.39	0.22	6.27	5.06	0.65	0.35	0.18	0.51	0.37	11.4	38.74	5604
2	22	0.29	0.19	0.12	1.93	3.15	0.51	0.14	0.12	0.24	0.54	11.2	33.68	1983
	43	0.47	0.3	0.17	3.38	3.97	0.58	0.26	0.15	0.38	0.41	10.9	36.2	3913
	81	0.63	0.39	0.23	5.91	4.82	0.63	0.35	0.18	0.5	0.37	11.6	36.99	5095
	100	0.65	0.41	0.23	6.6	5.2	0.66	0.37	0.18	0.52	0.36	11.8	38.43	5439
3	22	0.24	0.16	0.09	1.67	2.93	0.49	0.11	0.11	0.2	0.55	10.3	32.43	2366
	43	0.4	0.24	0.14	2.56	3.51	0.55	0.21	0.13	0.31	0.43	10.3	36.31	4013
	81	0.6	0.37	0.21	5.04	4.38	0.61	0.31	0.19	0.47	0.4	10.6	39.97	5653
	100	0.64	0.39	0.22	6.04	4.86	0.64	0.33	0.19	0.51	0.39	10.8	40.43	6052
Mean SE CV		0.49	0.31	0.17	4.16	4.13	0.59	0.26	0.16	0.39	0.43	10.9	36.57	4269
		0.05	0.03	0.01	0.55	0.24	0.02	0.03	0.01	0.04	0.02	0.15	0.76	414.5
		3.28	3.1	2.97	4.62	1.98	1.03	3.67	2.02	3.19	1.66	0.46	0.72	3.36

G^1/^: groups of genotypes; WR: water regimes (22, 43, 81, and 100% CET—WR1, WR2, WR3 and WR4, respectively); NDVI: normalized difference vegetation index; SAVI: soil-adjusted vegetation index; PRI: photochemical–physiological reflectance index; DVI: difference vegetation index; GRVI: green–red vegetation index; GNDVI: green normalized difference vegetation index; NDRE: normalized difference red edge; TCARI: chlorophyll absorption and reflectance index; OSAVI: optimized soil-adjusted vegetation index; TO: TCARI/OSAVI; GH: grain humidity (%); MTG: mass of thousand grains (g); GY; grain yield (Kg.ha^−1^). Group 1: Aliança, BRS 254, BRS 404, CPAC 01019, PF 020062, and PF 080492; Group 2: BR 18_Terena, MGS Brilhante, PF 020037, and PF 120337; Group 3: BRS 264, BRS 394, CPAC 01047, CPAC 07258, CPAC 8318, CPAC 9110, PF 100368, and TBIO Sintonia. SE: standard error; CV: coefficient of variation (%).

**Table 3 plants-12-03571-t003:** Mean values morphophysiological and drought tolerance index of wheat according to the groups of genotypes and water regimes in the 2018 and 2019 cropping years.

G^1/^	WR	Variables													
		WUE	Cha	Chb	RM	SM	*A*	*gs*	iWUE	Ci	*E*	Fv’/Fm’	ETR	Fv/Fm	DRI
1	22	22.7	30.9	7.2	4.9	10.4	8.7	0.09	121.9	188	2.5	0.46	122	0.8	1.01
	43	19.5	35.5	10.5	5	12.2	15.8	0.13	96.8	204	3.2	0.5	144	0.82	
	81	14.3	39.4	14.1	5.8	15.7	22.3	0.33	67.8	248	6.3	0.57	148	0.82	
	100	12.5	41.0	16.0	5.7	15.6	22.7	0.48	47.2	275	8.7	0.59	151	0.83	
2	22	20.3	32.0	8.2	5.4	11.1	9.7	0.09	125.6	193	2.5	0.47	133	0.81	0.95
	43	19.1	35.3	11.4	9.8	12.9	15.1	0.12	108.4	175	3.2	0.51	146	0.83	
	81	14.1	40.2	16.1	6.1	13.6	21.6	0.39	55.5	263	7.8	0.6	150	0.83	
	100	12.2	41.6	16.7	5.6	15.2	22.1	0.53	41.8	273	9.9	0.6	164	0.83	
3	22	22.0	27.4	5.4	4.8	9.9	8.1	0.09	127.1	168	2.6	0.4	99	0.8	0.93
	43	20.3	32.6	8.2	5.1	11.6	16.0	0.1	112.0	144	2.9	0.5	116	0.81	
	81	16.0	38.6	13	5.2	15.7	23.1	0.41	56.4	259.9	7.68	0.6	159	0.82	
	100	13.6	40.2	14.2	5.5	17.1	23.9	0.61	49.3	294.9	11.55	0.6	142	0.83	
Mean SE CV		17.2	36.1	11.7	5.7	13.4	17.4	0.28	84.1	224.2	5.75	0.5	140	0.82	0.96
		1.1	1.34	1.1	0.4	0.7	1.7	0.06	4.87	14.56	0.96	0.02	5.49	0.09	0.01
		2.2	1.28	3.3	2.3	1.8	3.4	7.05	1.12	2.25	5.77	1.29	1.36	0.13	0.11

G^1/^: group of genotypes; WR: water regimes (22, 43, 81, and 100% CET—WR1, WR2, WR3 and WR4, respectively); WUE: water-use efficiency; Cha: chlorophyll a; Chb: chlorophyll b; RM: root mass; SM: shoot mass; *A*: net assimilation of CO_2_ (µmol CO_2_ m^−2^ s^−1^); *gs*: stomatal conductance (mol H_2_O m^−2^ s^−1^); iWUE: intrinsic water-use efficiency (iWUE—A/gs: net assimilation of CO_2_/stomatal conductance); Ci: internal CO_2_ concentration (ppm); *E*: transpiration (mmol H_2_O m^−2^ s^−1^); Fv’/Fm’: effective quantum yield of photosystem II; ETR: electron transport rate; Fv/Fm: maximum quantum yield of photosystem II. DRI: drought resistance index. Group 1: Aliança, BRS 254, BRS 404, CPAC 01019, PF 020062, and PF 080492; Group 2: BR 18_Terena, MGS Brilhante, PF 020037, and PF 120337; Group 3: BRS 264, BRS 394, CPAC 01047, CPAC 07258, CPAC 8318, CPAC 9110, PF 100368, and TBIO Sintonia. SE: standard error; CV: coefficient of variation (%).

**Table 4 plants-12-03571-t004:** Relative importance (percentage) of variables for distance between genotypes according to Singh’s (1981) [29] criterion.

Variables	Percentage (%)
SAVI	33
OSAVI	33
DVI	11
NDRE	11
GNDVI	3
NDGI	2
RVI	2
NDVI	1
GRVI	1
GY	1
Other	<2

SAVI: soil-adjusted vegetation index; OSAVI: optimized soil-adjusted vegetation index; DVI: difference vegetation index; NDRE: normalized difference red edge; GNDVI: green normalized difference vegetation index; NDGI: normalized differential greenness index; RVI: ratio vegetation index; NDVI: normalized difference vegetation index; GRVI: green red vegetation index; GY: grain yield.

**Table 5 plants-12-03571-t005:** The regression equation for wheat variables as a function of water regime (*x*) for groups of genotypes.

Variable	Group of Genotype	Equation	R^2^
Main Coordinate 1 (latent variable)	1	y=−1.6339+0.0268x	0.98
	2	y=−1.4824+0.0266x	0.98
	3	y=−2.1757+0.0326x	0.98
Grain Yield (kg ha^−1^)	1	$y=5864.37/\left(1+\exp\left(-0.04*\left(x-31.3\right)\right)\right)$	0.99
	2	$y=5364.0/\left(1+\exp\left(-0.07*\left(x-29.36\right)\right)\right)$	0.99
	3	$y=6194.41/\left(1+\exp\left(-0.05*\left(x-31.32\right)\right)\right)$	0.99
Net CO_2_ Assimilation (*A*, µmol CO_2_ m^−2^ s^−1^)	1	$y=23.13/\left(1+\exp\left(-0.062*\left(x-30.24\right)\right)\right)$	0.99
	2	$y=23.18/\left(1+\exp\left(-0.047*\left(x-29.12\right)\right)\right)$	0.99
	3	$y=24.20/\left(1+\exp\left(-0.064*\left(x-32.53\right)\right)\right)$	0.99
Normalized Difference Vegetation Index (NDVI)	1	$y=-0.52+0.2562*\log\left(x\right)\;$	0.99
	2	$y=-0.46+0.2454*\log\left(x\right)$	0.99
	3	$y=-0.62+0.2739*\log\left(x\right)$	0.99
Water-use efficiency (WUE)	1	$y=25.37-0.131*\log\left(x\right)$	0.98
	2	$y=23.22-0.109*\log\left(x\right)$	0.99
	3	$y=24.65-0.108*\log\left(x\right)$	0.99
Intrinsic Water-use efficiency (iWUE)	1	$y=135.88-0.846*\log\left(x\right)$	0.96
	2	$y=133.23-0.833*\log\left(x\right)$	0.98
	3	$y=126.47-0.824*\log\left(x\right)$	0.97

Group 1: Aliança, BRS 254, BRS 404, CPAC 01019, PF 020062, and PF 080492; Group 2: BR 18_Terena, MGS Brilhante, PF 020037, and PF 120337; Group 3: BRS 264, BRS 394, CPAC 01047, CPAC 07258, CPAC 8318, CPAC 9110, PF 100368, and TBIO Sintonia. Water regime (WR1, WR2, WR3, and WR4—22, 43, 81, and 100% CET replacement, respectively).

## Data Availability

The data for this article can be shared upon reasonable request to the corresponding author.

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
