# Peer review of "Water Stress Alters Physiological, Spectral, and Agronomic Indexes of Wheat Genotypes"

_plants, 2023, doi:10.3390/plants12203571_

Round 1
Reviewer 1 Report
1. Is there a significant difference in DRI between group 1, Group 2 and group 3? The author should calculate the difference among them.
2. Table 4, What does authors mean by an abbreviation should be explained in the note.
3. The authors should present the sources, genotypes, and characteristics of the 18 different wheat used in the experiment. Why did the authors choose these 18 genotypes for the experiment?
4. what does mean “rainfed biotypes”, authors should define it. What is the difference if both rainfed and traditional types are planted in rainfed areas? What difference does it make if it's also grown in irrigated areas?
5. Rain-fed farming areas may have thicker clouds and less light. The authors may also need to consider the effects of low light conditions on wheat growth, or the absence of low light in that part of Brazil.
6. line 445-449, Please provide the source of the soil data, or describe the methods used to determine the soil data.
Author Response
Reviewer 1
- Is there a significant difference in DRI between group 1, Group 2 and group 3? The author should calculate the difference among them.
Authors: We calculated and included a figure with this variable.
- Table 4, What does authors mean by an abbreviation should be explained in the note.
Authors: We corrected and indicated the names of each symbol
- The authors should present the sources, genotypes, and characteristics of the 18 different wheat used in the experiment. Why did the authors choose these 18 genotypes for the experiment?
Authors: We included the source of genotypes.
- what does mean “rainfed biotypes”, authors should define it. What is the difference if both rainfed and traditional types are planted in rainfed areas? What difference does it make if it's also grown in irrigated areas?
Authors: We corrected the methods and discussion
- Rain-fed farming areas may have thicker clouds and less light. The authors may also need to consider the effects of low light conditions on wheat growth, or the absence of low light in that part of Brazil.
Authors: in the Brazilian Cerrado, there is no low light intensity in any period of the year.
- line 445-449, Please provide the source of the soil data, or describe the methods used to determine the soil data.
Authors: We collected the soil samples and analyzed
Reviewer 2 Report
In order to characterize wheat genotypes in relation to water deficit, the field experiments were conducted in 2018 and 2019 and 18 wheat genotypes were used. As a conclusion, the results showed the wheat genotypes can be selected by High-throughput and non-destructive method.
Some suggestions for modification are as follows:
1. The topic of the article was “Water stresses of …”, while in the experiment design and the result, only one of the content which was “water deficit or drought”.
2. In introduction, there is 12 paragraphs, but the core meaning was not clear, yet. Please refine the central idea and existing problems.
3. In the results, what’s the role of the data of table 2 and table 3?
4. Figure 1 showed the dendrogram of eighteen wheat genotypes. But what’s the method of classification according to?
5. Amount of abbreviation was used which could confused the readers. Please settle this problem.
Minor editing.
Author Response
Reviewer 2
In order to characterize wheat genotypes in relation to water deficit, the field experiments were conducted in 2018 and 2019 and 18 wheat genotypes were used. As a conclusion, the results showed the wheat genotypes can be selected by High-throughput and non-destructive method.
Some suggestions for modification are as follows:
1.The topic of the article was “Water stresses of …”, while in the experiment design and the result, only one of the content which was “water deficit or drought”.
Authors; In several parts of the paper, we discuss the impact of water stress in the groups of wheat genotypes
- In introduction, there is 12 paragraphs, but the core meaning was not clear, yet. Please refine the central idea and existing problems.
Authors: We corrected and removed some no important parts.
- In the results, what’s the role of the data of table 2 and table 3?
Authors: These tables show mean data from the variables
- Figure 1 showed the dendrogram of eighteen wheat genotypes. But what’s the method of classification according to?
Authors: The method is presented in the statistical analysis: “Genotypes were grouped based on Mahalanobis distance using Ward's method. The Mojena criterion [58] was used to define the cutoff point in the dendrogram, and the relative importance (proportion) of variables in the distance between genotypes was determined by the Singh [59] criterion”.
- Amount of abbreviation was used which could confused the readers. Please settle this problem.
Authors: it isn't easy to alter this part, as several indexes were evaluated simultaneously.
Round 2
Reviewer 1 Report
This manuscript could be published.
Reviewer 2 Report
All my concerns have been addressed. I recommend acceptance of this manuscript.